# Revisiting the NPcis mouse model: A new tool to model plexiform neurofibroma

**Camille Plante**[1�], **Teddy Mohamad**[1�], **Dhanushka Hewa Bostanthirige**[1�], **Michel Renaud**[1], **Harsimran Sidhu**[1], **Michel ElChoueiry**[1], **Jean-Paul Sabo Vatasescu**[1], **Mikael Poirier**[1], **Sameh Geha**[2,3,4], **Jean-Philippe Brosseau**[1,3,4]*

**1** Department of Biochemistry and Functional Genomic, Université de Sherbrooke, Sherbrooke, Quebec, Canada, **2** Department of Pathology, Université de Sherbrooke, Sherbrooke, Quebec, Canada, **3** Centre de recherche du Centre Hospitalier de Universitaire de Sherbrooke, Sherbrooke, Quebec, Canada, **4** Institut de recherche sur le Cancer de l'Université de Sherbrooke, Sherbrooke, Quebec, Canada

☉ These authors contributed equally to this work.
* jean-philippe.brosseau@USherbrooke.ca

**Data Availability Statement:** All relevant data are within the paper and its Supporting information files.

## Abstract

Neurofibromatosis Type I (NF1) is a rare genetic disorder. NF1 patients frequently develop a benign tumor in peripheral nerve plexuses called plexiform neurofibroma. In the past two decades, tissue-specific *Nf1* knockout mouse models were developed using commercially available tissue-specific Cre recombinase and the *Nf1* flox mice to mimic neurofibroma development. However, these models develop para-spinal neurofibroma, recapitulating a rare type of neurofibroma found in NF1 patients. The NPcis mouse model developed a malignant version of neurofibroma called malignant peripheral nerve sheath tumor (MPNST) within 3 to 6 months but intriguingly without apparent benign precursor lesion. Here, we revisited the NPcis model and discovered that about 20% display clinical signs similar to *Nf1* tissue-specific knockout mice models. However, a systematic histological analysis could not explain the clinical signs we observed although we noticed lesions reminiscent of a neurofibroma in a peripheral nerve, a cutaneous neurofibroma, and para-spinal neurofibroma on rare occasions in NPcis mice. We also observed that 10% of the mice developed a malignant peripheral nerve sheath tumor (MPNST) spontaneously, coinciding with their earring tag identification. Strikingly, half of the sciatic nerves from NPcis mice developed plexiform neurofibroma within 1–6 months when intentionally injured. Thus, we provided a procedure to turn the widely used NPcis sarcoma model into a model recapitulating plexiform neurofibroma.

## Introduction

Neurofibromatosis Type I (NF1) is a neurocutaneous genetic disease [1–3]. It is caused by the loss of function of the tumor suppressor gene encoding neurofibromin (i.e. the *NF1* gene) [4, 5]. Neurofibromin is able to attenuate Ras signaling due to its GTPase activity [6], and hence, *NF1* mutations inhibiting this activity promote tumor development through excessive Ras signaling. The cardinal features of NF1 are due to *NF1* inactivation in melanocytes (e.g. café-au-

**Funding:** Fonds de recherche du Québec - Santé. Grant # 281660. We would like to state that this funder had no role in study design, data collection and analysis, decision to publish, or preparation of the manuscript.

**Competing interests:** The authors have declared that no competing interests exist.

lait macules, iris hamartomas) or Schwann cells (neurofibromas). At least two subtypes of neurofibromas existed based on their different clinical behavior [1]. 99% of NF1 patients developed cutaneous neurofibromas (cNFs). They can have hundreds or thousands of cNFs covering their bodies. The onset is usually around puberty, and cNF has zero malignant potential [1]. In sharp contrast, plexiform neurofibroma have a penetrance of 25–50%. There is usually 1–2 pNF per patient. Their onset can be as early as congenital, and they risk transforming into a frank malignant tumor [malignant peripheral nerve sheath tumor (MPNST)]. 8–13% of NF1 patients will develop an MPNST [7]. Typically, the neurofibroma to MPNST sequence progress by the subsequent inactivation of the *CDKN2A* locus followed by a gene encoding a member of the PRC2 complex (e.g. *EED* or *SUZ12*) [8–11].

Several mice models were developed to better understand tumorigenesis in the context of this rare disease. It took about a decade from the time the *NF1* gene was first cloned to develop the first mouse model recapitulating neurofibroma and MPNSTs. *Nf1* knockout mice are not viable, and unlike humans, $Nf1^{+/-}$ mice do not develop any of the cardinal features of NF1 patients although they undergo loss of heterozygosity in other tissues to yield other NF1-related tumors such as pheochromocytomas [12]. Tissue-specific *Nf1* knockout using a genetically engineered mouse model expressing Cre recombinase under a Schwann cell promoter (e.g. *Krox20*-Cre and lox P sequence flanking *Nf1* exon 31) successfully inactivate *Nf1* expression in the Schwann cell lineage [13]. As a result, *Krox20*-Cre $Nf1^{f/-}$ mice develop neurofibroma with high penetrance. Since this seminal discovery, several Cre recombinases under distinct Schwann cell promoters have been used in combination with the *Nf1* flox mice, and produced similar results [14–19]. However, all but one generates exclusively para-spinal neurofibroma, a rare subtype found in NF1 patients. When *P0*-Cre $Nf1^{f/f}$ mice were aged, none of them developed neurofibroma [20]. Since a nerve injury may enhance the development of neurofibroma, it was hypothesized that *P0*-Cre $Nf1^{f/f}$ sciatic nerve injury may enhance neurofibroma development. Indeed, about a third of the mice developed *bona fide* plexiform neurofibroma within 6 to 8 months [20]. Thus, there is a need to develop neurofibroma models with rapid onset, high penetrance, and representing the main subtype found in NF1 patients: plexiform neurofibroma.

The most widely used mouse model for MPNST is the NPcis model [21]. *p53* mutations were reported in MPNST tumors [22]. Most NPcis mice die within 6 months due to the development of various sarcomas. MPNSTs developed within 3 to 6 months with 30% penetrance. Intriguingly there is no apparent benign precursor lesion. However, at the time when it was developed more than 20 years ago, no neurofibroma mice model existed. It was only realized later that the main manifestations of NF1 were either not recapitulate in mice (e.g. Lisch nodules, CALMs) or recapitulate only a rare subtype (e.g. para-spinal neurofibroma [13–19], diffuse cNF [14, 18]). To develop a novel mouse model for plexiform neurofibroma, we decide to revisit the NPcis model. In doing so, we optimize this model to yield a rapid and high penetrance plexiform neurofibroma model.

## Methods

### Mice

All mice were housed in the Animal Care Facility at the Université de Sherbrooke. We follow the principles of laboratory animal care (NIH publication No. 86–23, revised 1985). All procedures were approved by the Animal Research Ethics Committee of the Université de Sherbrooke in accordance with the Canadian Council on Animal Care standards.

All mouse strains were maintained in a room with 12/12 (day/night) light cycle with a temperature of 70-72˚F. NPcis and Hoxb7-cre $Nf1^{f/f}$ mice were rederived on C57bl/6 background.

Genotyping was performed by PCR as reported elsewhere [14, 21]. The investigators were not blinded to the group allocation during the experiment. No mice were excluded for any reason. When a point limit or end of the experiment is reached, mice are euthanized using isoflurane 5% mixed with oxygen (1–2 L/min), followed by asphyxiation using $CO_2$. Tissues are collected and whole spinal cord dissection is performed as described elsewhere [23].

To perform sciatic nerve-induced injury, one-month-old mice were anesthetized by isoflurane 5% mixed with oxygen (1–2 L/min) and subsequently placed on a warmed surgery plate maintained around 85˚F and mounted with a mask to maintain anesthesia (2% isoflurane). A cream is applied to the eyes to prevent dryness. The lower back is shaved and subsequently depilated using the NAIR™ cream. Excess is removed after 1–2 min.

A skin incision was made above the sciatic nerve using extra fine Bonn scissors (#14084–08). Upon locating the sciatic nerve in the quadricep muscle cavity with a fine curved twizzer (#91117–10). For the "needle" strategy, a 27G needle was used to perforate the sciatic nerve and disrupt the nerve fibers longitudinally in a few bundles. For the "cut" strategy, the peroneal and tibial nerve were cut at the sciatic nerve ramification (sparing the sural nerve) using extra fine Bonn scissors (#14084–08). The sciatic nerve was placed back in the muscle cavity. The skin was sutured using monocryl 4–0 wire (adsorbable).

At the end of the surgery, the mask is removed, and the mouse is allowed to recover on the warmed plate until the mouse is ambulant. Mice are closely follow-up for tumor formation.

## Histological characterization

Tissues were fixed in 10% formalin-buffered solution for at least 48 hours and subsequently washed with EtOH 70%. Then all samples were transferred to the Histology Research Core of the FMSS at the Université de Sherbrooke, for their circulation and inclusion. Briefly, tissues were paraffin-embedded (Thermo Fischer Varistain or Citadel 2000), were sectioned at 5 μm using a microtome (Leica Histocore Multicut) and were allowed to dry on Diamond white glass microscope slides (Globe Scientific; #1358W) at room temperature (r.t.). Hematoxylin and eosin (H&E) staining was performed using Gill 3 hematoxylin (Thermo Scientific, 72604), followed by short washes with acidic water and ammoniacal water and eosin-Y in 1% alcoholic solution (Fischer Scientific, #245827) as counterstaining. Mast cell staining was performed using a solution of 0,1% toluidine blue followed by overnight air drying. For immunohistochemistry, the Vecta Stain Elite ABC kit (Vectorlabs, PK-6100) was used according to the manufacturer's protocol. The following primary antibodies were used in the immunohistochemistry studies: S100 (Agilent Technologies, GA-50461-2); desmin (abcam, ab15200); SMA (GeneTex, GTX100034); Nf1 (Santa Cruz, sc-376886); p53 (Santa Cruz, sc-126); Iba1 (Fujifilm, 019–19741). The pathologist (S. Geha) reviewed the histologies. For immunohistochemistries in "Fig 4c", slides were treated with a solution of 3% $H_2O_2$ in NaOH 0.5N for 20 min to depigment the skin.

## qPCR

qPCR was performed using gene-specific primers and cDNA from RNA extracted from FFPE (formalin-fixed paraffin-embedded) histology slides based on the RNeasy FFPE kit (Qiagen; 73504). Briefly, a scalpel was used to grossly dissect the tissue of interest from FFPE slides (3 slides per tissue/mouse were pooled for each replicate) and process following the manufacturer's recommendation using xylene as the deparaffinization solution, proteinase K in Tris Guanidinium thiocyanate 4M as the lysis solution (15 min, 55˚C, then 15 min, 80˚C) and $NH_4Cl$ 5M as the demodification solution (20 min, 95˚C) as previously done [24]. The resulting column-based purified total RNA extract was quantified using Nanodrop and 150 ng from each

sample were submitted to a reverse transcription reaction using Roche Transcriptor™ and N6 random primers. Finally, cDNA was diluted with 440 uL of RNase DNase free water and used as template for a qPCR reaction using primers specific for *Nf1* (Nf1_F 5`–GCAACTTGCCACTCCCTACTGA–3`; Nf1_R 5`– ATGCTGTTCTGAGGGAAACGCT–3`); *p53* (p53_F 5`–CTGCACTTGGATCGGGAAGT–3`; p53_R 5`–GCCACGAAAACAAAGTCCCC–3`) and *GAPDH* (GAPDH_F 5`–TGACGTGCCGCCTGGAGAAA–3`; GAPDH_R 5`– TGACGTGCCGCCTGGAGAAA–3`) as housekeeping gene in the 2X SyBr Green mix buffer (Quantabio, 95054-02K) under the following cycling conditions: 95˚C, 3min; [(95˚C, 15sec, 60˚C, 30sec, 72˚C, 30sec) X 50], 72˚C, 5min. Relative expression was calculated using qBase [25].

### Statistical analyses

A Fischer exact t-test "Figs 1d and 5b" or a one-way ANOVA "Fig 7c" was performed to determine if the two groups were different (P = 0.05 or below).

## Results

We generated and aged a cohort of NPcis (n = 61) and their wild-type littermates (n = 56) "S1 Table". 35 out of 61 (57%) developed at least one solid mass suspected as sarcoma within 6 months "Fig 1a", "S1 Table". The majority of these masses were sub-cutaneous. Very few were found in the dermis (cutaneous) or abdominal cavity "Fig 1b", "S1 Table". Around two-third of the masses were located in the trunk, and around one-third in the limbs. Only one was found in the head and neck area "Fig 1c", "S1 Table". Intriguingly, the mass incidence was

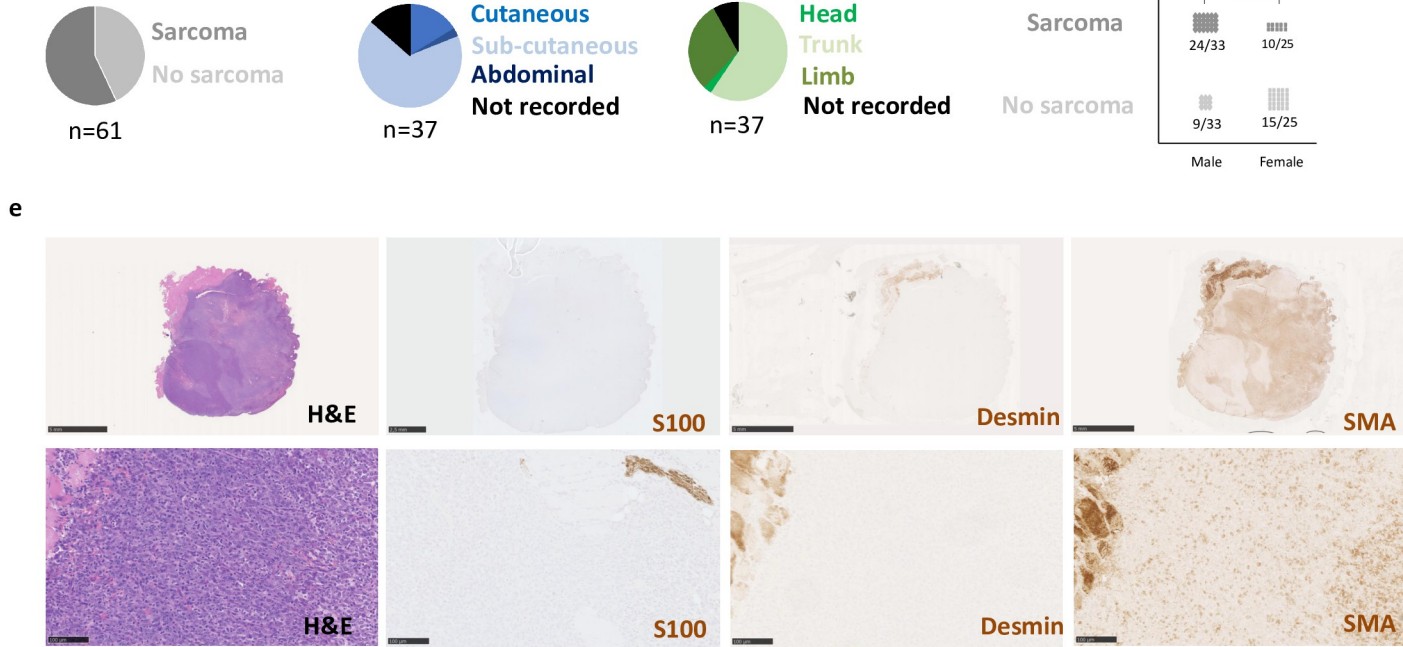

**Fig 1. NPcis derived sarcoma penetrance and tissue distribution. a-c** Pie chart representing the **a** percentage of mice developing sarcomas, **b** the sarcoma tissue sub-location and **c** the sarcoma body distribution. **d** Sarcoma incidence in relation to mouse sex. NPcis males [24/33 (72%)] develop more frequently sarcoma than NPcis females [10/15 (40%)]. **e** Representative H&E and S100, desmin and SMA immunohistological sarcoma characterization from NPcis mice in low (upper panels, scale bar equals to 5 mm) and high (bottom panels, scale bar equals to 100 µm) magnification.

more elevated in males than in females "Fig 1d", "S1 Table". In addition to MPNST, the NPcis mouse model is known to develop other types of sarcomas such as leiomyosarcoma and rhabdomyosarcoma [21]. These sarcomas with muscle differentiation are usually positive for muscle markers such as smooth muscle actin (SMA) and/or desmin. In contrast, most MPNSTs should show some S100 expression and be negative for muscle markers [26]. To determine the MPNST over the non-MPNST sarcomas, we harvested and performed a thorough histological evaluation on 26 tumors (harvested and kept for histology) out of 36 tumors (identified at necropsy) using S100, desmin and SMA as markers "Fig 1e and S1 Fig". Surprisingly, we found few tumor cells expressing S100 although S100 marks normal nerve within the sarcomas "Fig 1e and S1 Fig". Moreover, most masses show positivity for SMA but very little for desmin, although there is consistent staining in the muscle cells frequently surrounding or infiltrated by the sarcomas "Fig 1e and S1 Fig". Overall, it indicates that the sarcomas spontaneously developing in the NPcis mice are histologically different from human MPNST [27].

The NPcis mouse model is known to develop MPNSTs without apparent precursor lesions (neurofibroma). Surprisingly, we observed that around 20% of NPcis mice manifest one or more phenotypes commonly used as endpoints for indicating neurofibroma development in *Nf1* tissue-specific knockout mouse models "Table 1 and S1 Table". Those clinical signs are either related to the general health status of the mouse (low activity, ruffled fur, thinness) or are considered to be more specific for para-spinal neurofibroma development [kyphosis (hunched posture), ambulation difficulties, lower limb paralysis] [28, 29]. The Table 1 lists the NPcis mice manifesting the aforementioned phenotypes. None of the 13 mice listed in Table 1 display complete paralysis of the lower limb nor ruffled fur has witnessed in *Nf1* tissue-specific knockout mice models of neurofibroma [14, 16]. Interestingly, a minority (2 out of 13) developed a sarcoma "Fig 2a" and "S1 Table". To identify any abnormalities that could explain the clinical signs observed, these 13 mice were submitted to a complete peripheral nerve dissection [23] and histological evaluation. None of the dorsal root ganglions inspected and measured were above 1 mm in diameter, a measure used to gauge the presence of para-spinal neurofibroma in mice models [14]. Therefore, we submitted the six most enlarged dorsal root ganglions for histology and compared them with a para-spinal neurofibroma from the *Hoxb7*-Cre *Nf1* [f/f], an established neurofibroma model [14]. We could identify one dorsal root ganglion

**Table 1. Neurofibroma-like clinical signs noticed in NPcis mice.**

| Mouse # | Sex | Para-spinal neurofibroma | | | Cutaneous neurofibroma | General Health | | |
|---------|-----|--------------------------|---|---|------------------------|----------------|---|---|
| | | Hunched posture | Ambulation difficulties | Lower limb paralysis | cNF | Low activity | Thinness | Ruffled fur |
| 46312 | M | | √ | | | | | |
| 46513 | F | √ | | | | | | |
| 46526 | M | | √ | | | | | |
| 46567 | M | √ | √ | | | | | |
| 46601 | M | | √ | | | | | |
| 46605 | F | √ | | | | √ | | |
| 46608 | F | √ | √ | | | √ | | |
| 46740 | F | | | | | √ | √ | |
| 46768 | F | | | | | | √ | |
| 46864 | F | | | | | √ | √ | |
| 46882 | M | √ | | | | √ | √ | |
| 46886 | F | | | | √ | | | |
| 46901 | F | √ | | | | √ | | |

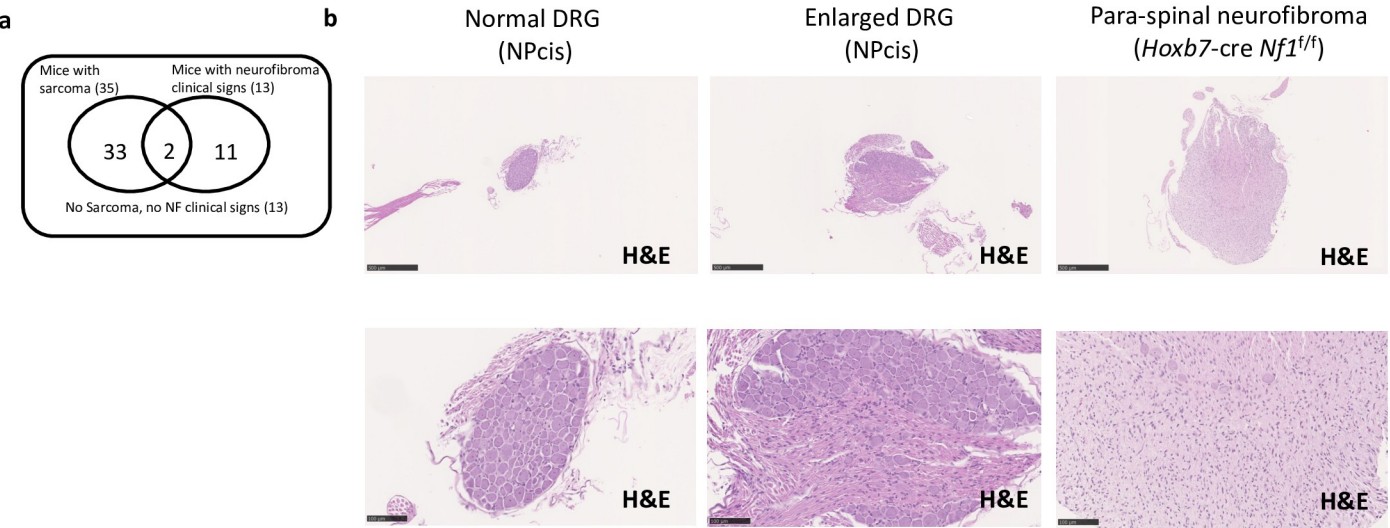

**Fig 2. Histological characterization of dorsal root ganglion of NPcis mice. a** Venn diagram representing the low overlap between the NPcis mice that develop a sarcoma and the one that manifests NF clinical signs **b** low (upper, scale bar equals 500 μm) and high (bottom, scale bar equals 100 μm) magnification H&E of a normal dorsal root ganglion (left), an enlarged dorsal root ganglion (middle) of a NPcis mouse and an enlarged dorsal root ganglion (right) of a *Hoxb7*-Cre *Nf1*^f/f mouse.

transitioning into a neurofibroma "Fig 2b". We conclude that spontaneous para-spinal neurofibroma in NPcis mice is a very rare event.

Next, we scrutinized the peripheral nerves of the 13 mice listed in Table 1. All peripheral nerve tissues appeared normal except in one case. We noticed a nerve with an enlarged region "Fig 3a". Histological evaluation by H&E revealed the presence of a plexiform neurofibroma "Fig 3b". This is supported by the S100 (Schwann cells) signal fading in the neurofibroma area. As the presence of mast cells is a hallmark of neurofibroma [30], we also performed toluidine blue (mast cells) staining but could not detect a high infiltration of mast cells. Of note, mast cells are not essential for pNF development [16, 31]. We conclude that although rare, NPcis mice can develop neurofibroma in a peripheral nerve, which is common in NF1 patients but rare in *Nf1* tissue-specific knockout mouse models of neurofibroma [20].

Interestingly, we also examined the skin of NPcis mice (n = 61) and in one case, we noticed a lesion in the neck area "Fig 4a and 4b" Of note, this is the very place where cNF develops in the *Hoxb7*-Cre *Nf1*^f/f [14] and *Prss56*-Cre *Nf1*^f/f [18] mouse models. To confirm it is cNF, we performed histological analysis "Fig 4c". The results indicate that although rare, NPcis mice can develop skin lesion mimicking cNF.

Intriguingly, we also noticed that 10% of mice developed an ear mass "Fig 5a and 5b", "S1 Table". Histological evaluation revealed that these masses were MPNSTs as in "Figs 1, 5c and 5d". Since these MPNSTs coincide with their earring, we hypothesize that the wound-induced earring enhances the likelihood of tumor formation at the exact wound location. It is known that a nerve injury can enhance the development of neurofibroma [20], but it was never attempted on the NPcis mice. Therefore, we tried a procedure where we intentionally injured the sciatic nerves of NPcis mice to enhance the likelihood of plexiform neurofibroma formation in peripheral nerves. In this sense, we proceed with sciatic nerve injuries in 1-month old NPcis (n = 24 sciatic nerves) and their wild-type littermates (n = 8 sciatic nerves). After 3 months, we euthanized the mice, harvested the sciatic nerves and proceeded with histological evaluation. The results indicate that 17% (4 out of 24) sciatic nerves progress into plexiform

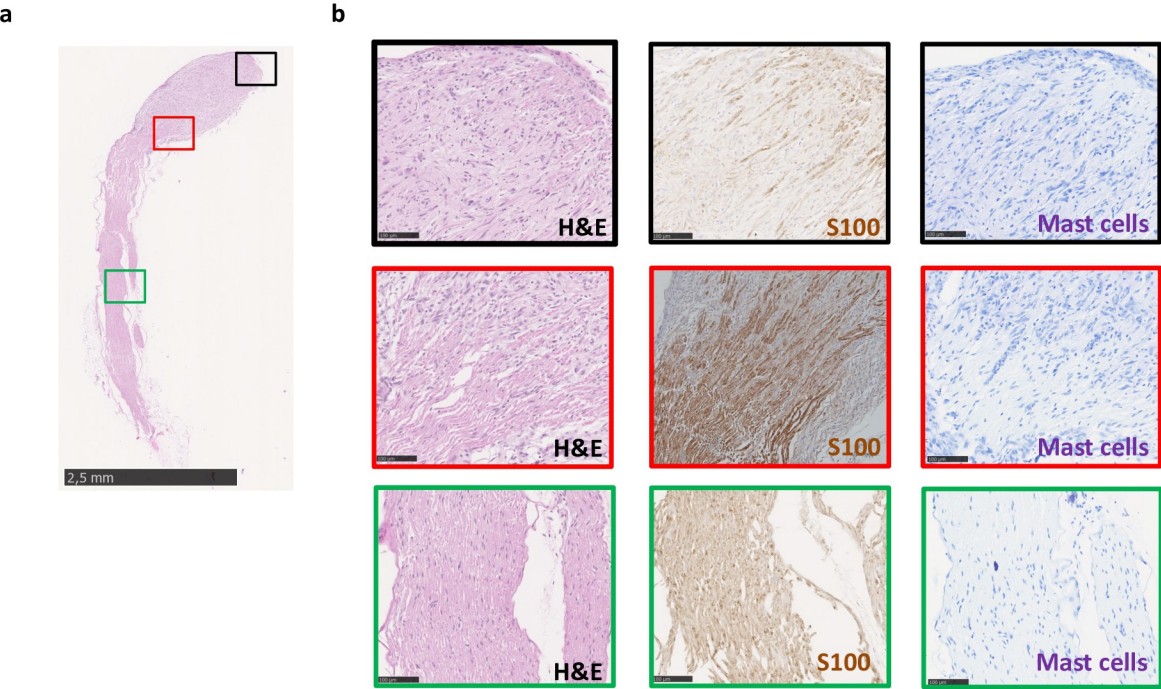

**Fig 3. Histological characterization of a plexiform neurofibroma arising in a peripheral nerve of a NPcis mouse. a** low magnification (scale bar equals to 2.5 mm) H&E of an enlarged peripheral nerve of a NPcis mouse **b** Histological characterization by H&E (left), immunostaining for S100 [middle (Schwann cells)] and toluidine blue [right (mast cells)]. Scale bar is equal to 100 μm.

neurofibroma while none (0 out of 8) of the wild-type sciatic nerves do so "Fig 6a–6e and S2 Fig", "S2 Table". We conclude that intentional sciatic nerve injuries in NPcis mice yield plexiform neurofibroma although at low penetrance.

The procedure we used to damage the nerves is very similar to the one Chen et al [29] used for their allograft. The concept is to disrupt the sciatic nerve fibers (e.g. with a needle) and to create a physical space for the tumoral cells to infiltrate the nerve. To improve the yield (plexiform neurofibroma incidence) of the NPcis mice nerve injury model, we decided to perform a more severe damage by cutting the peroneal and tibial nerve at the sciatic nerve ramification (sparing the sural nerve) "Fig 6f". In an independent cohort, 3 NPcis mice (n = 6 sciatic nerves) were injured as in "Fig 6a–6e", (referred as the needle method) and 9 NPcis mice (n = 18 sciatic nerves) were injured using the aforementioned method (referred as the cut method). To ensure that the mice have sufficient time to develop plexiform neurofibroma, they were aged up to 6 months whenever possible "S3 Table". Strikingly, 50% (9 out of 18) of sciatic nerves injured using the cut method develop a plexiform neurofibroma within 1 to 6 months "Fig 6g and S3 Fig", "S3 Table". Of note, we could not discriminate the pNF by histology based on the injury method used "S4 Fig". To complement the histological characterization of pNF from injury-induced sciatic nerve, we performed immunostaining using a macrophage marker. As expected, we noticed positive staining in pNF but minimal staining in the injured sciatic nerves that did not develop pNF "S5 Fig". Overall, plexiform neurofibroma in peripheral nerves is a rare event in *Nf1* tissue-specific knockout mouse models and NPcis mouse models but can be significantly enhanced by cutting the peroneal and tibial nerve of NPcis mice.

On one hand, it was demonstrated that spontaneous sarcoma from NPcis mice undergo loss of heterozygosity (Nf1 and p53) in the malignant Schwann cells [21]. On the other hand,

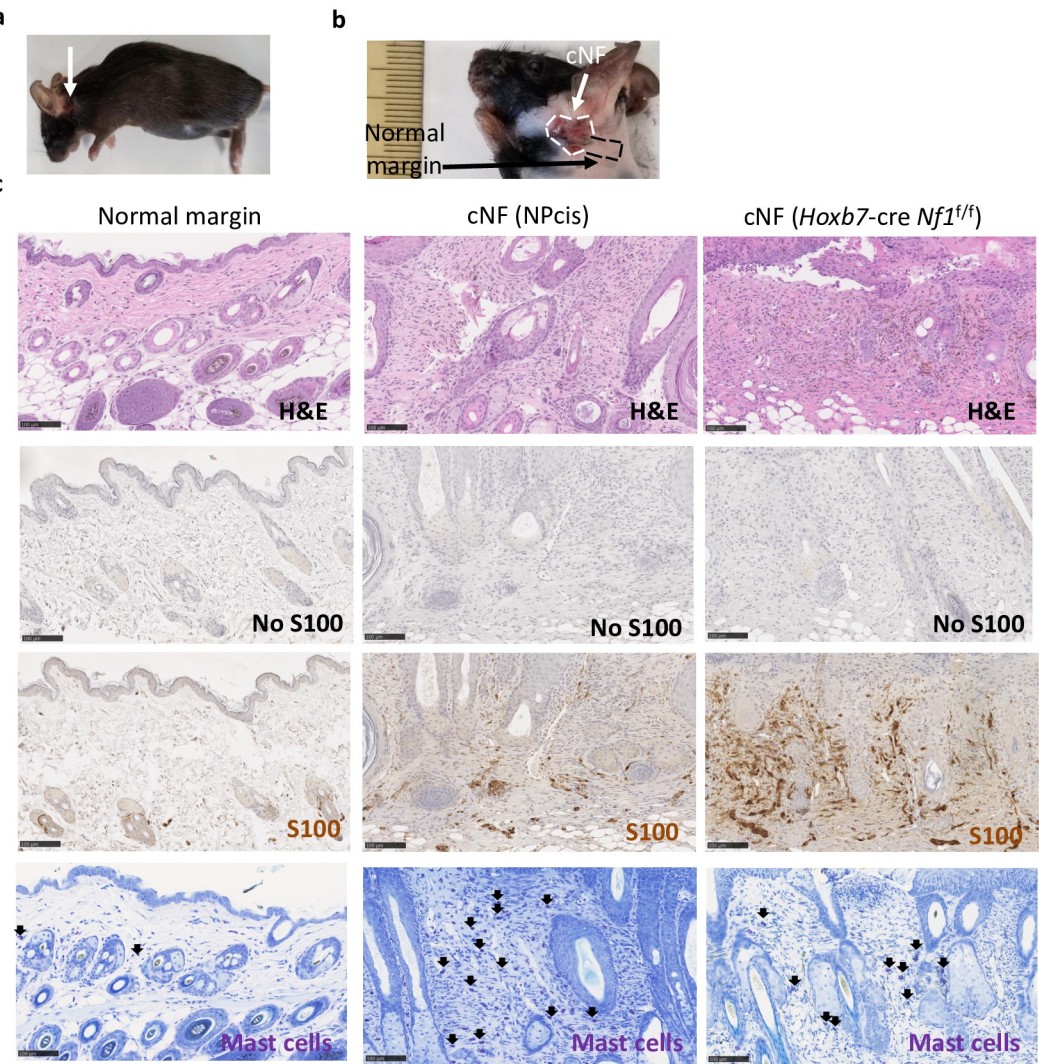

**Fig 4. Histological characterization of a cutaneous neurofibroma arising in a NPcis mouse. a** Gross picture of a mouse developing a cNF **b** Gross picture depicting the cNF and its normal margin harvested for histological evaluation. **c** Histology of the normal margin skin of a NPcis mouse (left), skin lesion from a NPcis mouse (middle) and a skin lesion from a *Hoxb7*-Cre *Nf1*^f/f^ mouse (right). Histological characterization by H&E (upper), immunostaining for S100 [middle (Schwann cells)] and toluidine blue [bottom (mast cells)]. Arrows point to the mast cells. Black scale bar is equal to 100 µm.

*Nf1* loss of heterozygosity in Schwann cells is required for pNF development. Therefore, we evaluate the status of *Nf1* and *p53* in our injury-induced pNF model. To do so, we performed immunostaining using NF1 and p53 antibodies on normal sciatic nerves (no injury) from wild-type mice (n = 8), injury-induced sciatic nerves from NPcis mice that develop pNF (n = 12), injury-induced sciatic nerves from NPcis mice that did not develop (n = 9) pNF and spontaneous sarcoma from NPcis mice (n = 23) "Fig 7a". Then, the staining intensity was scored and output as a bar graph representing the percentage of tissues assigned to each staining intensity level. The results indicate that NF1 and p53 expression are largely preserved in non-malignant tissues whereas it is lowly expressed in malignant tissues as expected "Fig 7b". We also performed qPCR using *Nf1* and *p53* specific primers on normal sciatic nerves (no injury) from wild-type mice (n = 2), injury-induced sciatic nerves from NPcis mice that

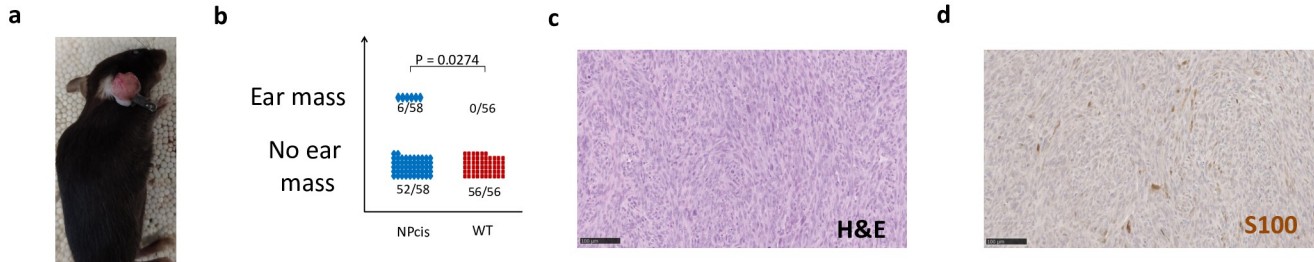

**Fig 5. 10% of NPcis develop ear mass coinciding with earring tag. a** Picture of a NPcis mouse bearing an ear mass coinciding with the earring tag. **b** Ear mass incidence in relation to mice genotype. A fraction of NPcis mice [6 out of 58 (10%)] developed at least one ear mass but none of the 56 wild-type littermates did. **c** Representative H&E of an ear mass from one of the NPcis mouse. **d** Histological characterization by immunostaining for S100 (Schwann cells).

**Fig 6. Nerve injury in the sciatic nerves of NPcis mice induces the development of plexiform neurofibroma. a** Picture of a WT mouse 3 months post sciatic nerve injury by needle. **b** Histological characterization by H&E (left), immunostaining for S100 [left (Schwann cells)] and toluidine blue [right (mast cells)]. Arrows point to the mast cells. Black scale bar is equal to 100 μm. **c** Picture of a NPcis mouse 3 months post sciatic nerve injury by needle. **d** Histological characterization by H&E (left), immunostaining for S100 [left (Schwann cells)] and toluidine blue [right (mast cells)]. Black scale bar is equal to 100 μm. Malignant spindle cell proliferation with nuclear atypical and mitosis and/or necrosis distinguish sarcomas from plexiform neurofibroma. **e** Stacked histogram summarizing the percentage of normal, hypercellular, plexiform neurofibroma and MPNST in WT and NPcis following nerve injury by needle after 3 months in WT and NPcis mice. **f** Picture of a NPcis mouse submitted to surgery cutting the sciatic nerve at the sural, tibial and peroneal nerve ramification (sparing the sural nerve). **g** Stacked histogram summarizing the percentage of normal, hypercellular, pNF and MPNST in NPcis following nerve injury by needle or cut (at the sural, tibial and peroneal nerve ramification and sparing the sural nerve) after 1–6 months.

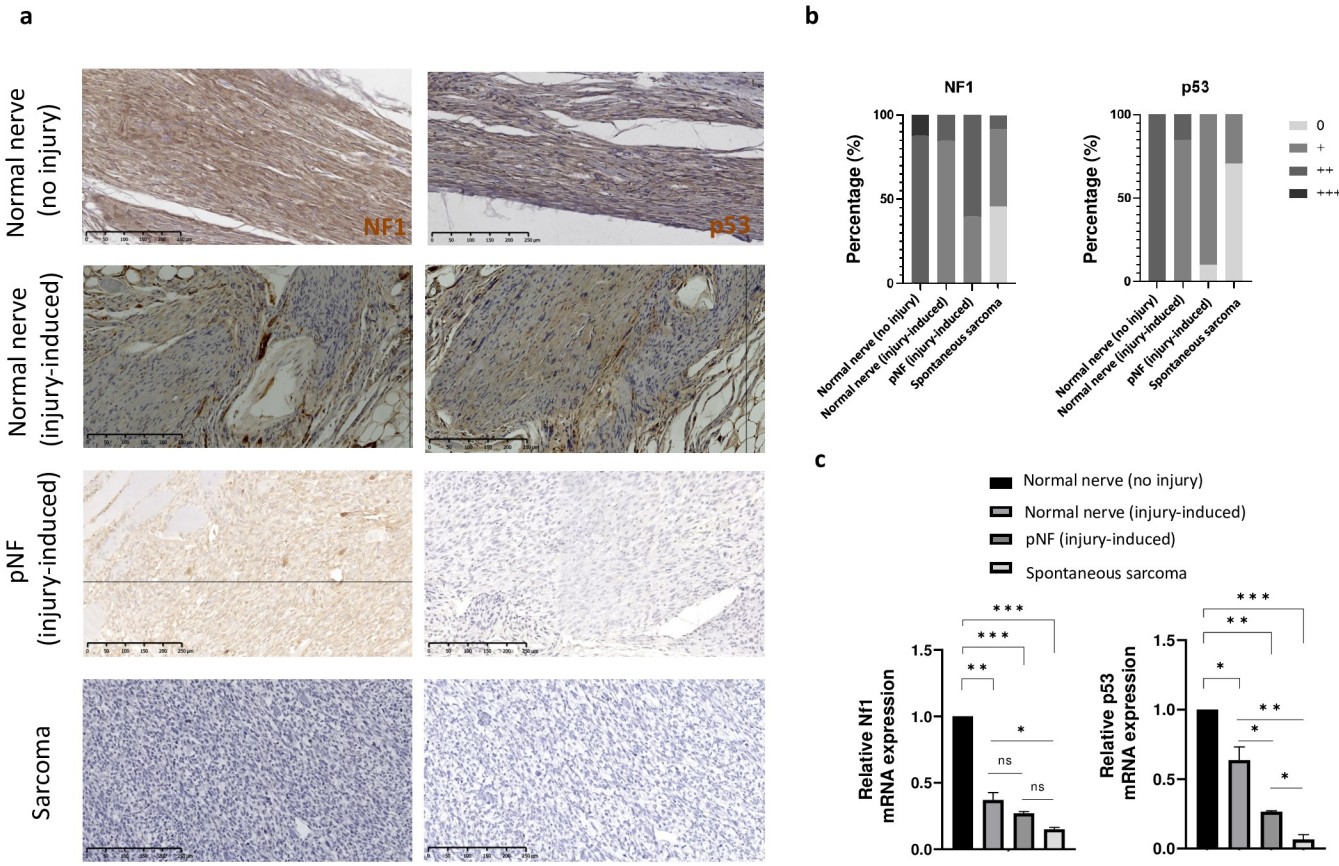

**Fig 7. NF1 and p53 expression in NPcis sciatic nerves injury-induced plexiform neurofibroma. a** Representative immunostaining for NF1 and p53 normal sciatic nerves (no injury) from wild-type mice, injury-induced sciatic nerves from NPcis mice that develop pNF, injury-induced sciatic nerves from NPcis mice that did not develop pNF and spontaneous sarcoma from NPcis mice. Scale bar is equal to 250 μm **b** Stacked histogram summarizing the percentage of tissues [normal sciatic nerves (no injury) from wild-type mice (n = 8), injury-induced sciatic nerves from NPcis mice that develop pNF (n = 12), injury-induced sciatic nerves from NPcis mice that did not develop (n = 9) pNF and spontaneous sarcoma from NPcis mice (n = 23)] categorized into different staining intensity level. **c** Bar graph representing the relative *Nf1* (left) and *p53* (right) gene expression level in normal sciatic nerves (no injury) from wild-type mice (n = 2), injury-induced sciatic nerves from NPcis mice that develop pNF (n = 2), injury-induced sciatic nerves from NPcis mice that did not develop (n = 2) pNF and spontaneous sarcoma from NPcis mice (n = 2).

develop pNF (n = 2), injury-induced sciatic nerves from NPcis mice that did not develop (n = 2) pNF and spontaneous sarcoma from NPcis mice (n = 2). Again, the results indicate that *Nf1* and *p53* expression are largely preserved in non-malignant tissues. More precisely, Nf1 expression is not significantly different between NPcis nerves and pNF but it is for p53. As expected, both Nf1 and p53 expression is significantly decreased in sarcoma compared to normal nerve from NPcis mice "Fig 7c". Thus, it suggests that loss of heterozygosity of *Nf1* and *p53* did not took place in injury-induced pNF although alternative strategies to lower p53 expression may exist.

## Discussion

Recapitulating human disease in mouse models is a critical milestone to advance drug development. In the context of NF1-associated tumors, the majority of the current published mouse models are limited to the recapitulation of rare sub-types of neurofibromas [13–19]. This may explain at least in part the discrepancies between the many successful drug target validation in

NF1 mouse models and the largely unsuccessful clinical trial targeting NF1-related tumors [32]. Here, we demonstrated that induced injuries in sciatic nerves of the NPcis mice promote the development of plexiform neurofibroma in peripheral injured nerves, the main form found in NF1 patients.

Trauma has been suggested as a possible cause of solitary neurofibroma development [33, 34]. It has been observed that the freckling of NF1 patients often coincides with body location where skin rubs against skin and where clothing is tight and rubs physically [35]. So far, nerve injury as a causative factor in neurofibroma development is anecdotal. In *P0*-Cre *Nf1*$^{f/f}$ mice, nerve crush is necessary for plexiform neurofibroma to develop at the site of injury [20]. In *Nf1*$^{+/-}$ mice, nerve injury (crush or cut) to the sciatic nerve is not sufficient to promote plexiform neurofibroma in most cases [36]. In the latter case, it is uncertain that *Nf1*$^{+/-}$ Schwann cells undergo loss-of-heterozygosity as it was not verified. Here, two approaches were used for sciatic nerve injury. The first strategy we used consisted of disrupting nerve fibers longitudinally. This procedure is necessary when performing Schwann cell graft [29] most probably because it creates physical space for the cells to integrate the nerve but also stimulates repair from the host nerve cells. This strategy yields 17% plexiform neurofibroma penetrance after 3 months "Fig 6e", and an additional 3 months did not increase the yield "Fig 6g". Of note, mice undergoing this surgery frequently show an inflamed heel of the related limb within a few days due to an alteration in gait. This observation was never reported when the same procedure was performed in immunocompromised mice graft [29], suggesting the contribution of the immune system to the inflamed heel phenotype.

The second strategy consists of cutting the sciatic nerve. A lesion to the sciatic nerve of a rodent is an established model of neuropathic pain [37]. Here, we performed a partial cut of the sciatic nerve, sparing the sural nerve ramification. Strikingly, 50% of the NPcis mice submitted to this strategy developed plexiform neurofibroma at the site of injury within 1 to 6 months. The fact that one mouse developed plexiform neurofibroma on both sciatic nerves only one month after the surgery and that the penetrance seems to plateau at 3 months using the needle method suggests that much less than 6 months may be sufficient for neurofibroma development in this injury-induce model using the cut method.

This model has several advantages over the current tissue-specific *Nf1* knockout models "Table 2". Since the pNF developed at the sciatic nerve injury site, it bypasses the tedious and lengthy whole peripheral nervous system dissection [23] normally required to isolate the ganglions where para-spinal neurofibroma develops in tissue-specific *Nf1* knockout models. In the later model, para-spinal neurofibroma usually develops within 6 to 12 months. Here, plexiform neurofibroma develops in 1 to 6 months in the injury-induced NPcis mice. Tissue-specific *Nf1* knockout models require the purchase/acquisition and breeding of two transgenic mice (i.e. the *Nf1* flox strain and a Schwann cell Cre strain), whereas the NPcis mice are directly available.

**Table 2. Comparison between the sciatic nerve injury-induced NPcis mice and *Nf1* tissue-specific knockout models.**

| Characteristics | Injury-induced NPcis | *Nf1* tissue-specific knockout | *P0*-Cre *Nf1*$^{f/f}$ |
|---|---|---|---|
| Time to develop neurofibroma | 1–6 months | 6–12 months | 6–8 months |
| penetrance | 50% | Close to 100% | 30% |
| Require highly qualified personnel for dissection? | No | Yes | No |
| Model accessibility | commercial | Require intense breeding | Require intense breeding |

## Conclusion

Here, we revisit the NPcis model and discovered that about 20% display clinical signs similar to pNF mouse models. However, a systematic histological analysis could not explain the clinical signs in most cases. As opposed to the current neurofibroma models that develop the paraspinal subtype, performing a severe damage to the sciatic nerves of a NPcis mouse yield a *bona fide* plexiform neurofibroma in a peripheral nerve.

## Supporting information

**S1 Fig. Histological characterization of spontaneous sarcoma from the NPcis model.** Full histological characterization (H&E and S100, desmin and SMA immunostaining) of sarcoma from NPcis mouse model.
(PDF)

**S2 Fig. Histological characterization of injury-induced sciatic nerve from "Fig 6e".** Full histological characterization (H&E) of injury-induced sciatic nerve from the NPcis mouse model.
(PDF)

**S3 Fig. Histological characterization of injury-induced sciatic nerve from "Fig 6g".** Full histological characterization (H&E) of injury-induced sciatic nerve from the NPcis mouse model comparing the needle and the cut method.
(PDF)

**S4 Fig. H&E needle vs cut injury-induced sciatic nerve.** NF1 and p53 immunostaining of spontaneous sarcoma from the NPcis mouse model.
(PDF)

**S5 Fig. Iba1 (macrophages) IHC of injury-induced sciatic nerves.** Iba1 immunostaining of injury-induced sciatic nerve from the NPcis mouse model.
(PDF)

**S6 Fig. Nf1 and p53 IHC of injury-induced sciatic nerve.** Nf1 and p53 immunostaining of injury-induced sciatic nerve from the NPcis mouse model.
(PDF)

**S7 Fig. Nf1 and p53 IHC of spontaneous sarcoma from the NPcis model.** NF1 and p53 immunostaining of spontaneous sarcoma from the NPcis mouse model.
(PDF)

**S1 Table. Details of the mice cohort from "Figs 1–5".**
(PDF)

**S2 Table. Details of the mice cohort from "Fig 6e".**
(PDF)

**S3 Table. Details of the mice cohort from "Fig 6g".**
(PDF)

## Acknowledgments

We are grateful to the JPBrosseau lab members for proof-reading the manuscript. We thank the Electron Microscopy and Histology Research Core of the Faculté de Médecine et des Sciences de la Santé at the Université de Sherbrooke for their histology. We thank the RNomic platform for their PCR services. The NPcis mice were a kind gift of Luis Parada (MSK, New

York) through the Lu Le lab (UTSW, Dallas). The Hoxb7-cre *Nf1* <sup>f/f</sup> mice were a kind gift of the Lu Le lab (UTSW, Dallas).

## Author Contributions

**Conceptualization:** Jean-Philippe Brosseau.

**Formal analysis:** Sameh Geha, Jean-Philippe Brosseau.

**Funding acquisition:** Jean-Philippe Brosseau.

**Investigation:** Camille Plante, Teddy Mohamad, Dhanushka Hewa Bostanthirige, Michel Renaud, Harsimran Sidhu, Michel ElChoueiry, Jean-Paul Sabo Vatasescu, Mikael Poirier, Jean-Philippe Brosseau.

**Methodology:** Dhanushka Hewa Bostanthirige, Jean-Philippe Brosseau.

**Project administration:** Jean-Philippe Brosseau.

**Supervision:** Jean-Philippe Brosseau.

**Visualization:** Jean-Philippe Brosseau.

**Writing – original draft:** Jean-Philippe Brosseau.

**Writing – review & editing:** Jean-Philippe Brosseau.

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
