## [Decision Letter · Decision Letter 0]

12 Dec 2023

PONE-D-23-19723Revisiting the NPcis mouse model: a new tool to model plexiform neurofibromaPLOS ONE

Dear Dr. Brosseau,

Thank you for submitting your manuscript to PLOS ONE. After careful consideration, we feel that it has merit but does not fully meet PLOS ONE’s publication criteria as it currently stands. Therefore, we invite you to submit a revised version of the manuscript that addresses the points raised during the review process.

We look forward to receiving your revised manuscript.

Kind regards,

Giulio Piluso, M.Sc.

Academic Editor

PLOS ONE

Journal Requirements:

"Fonds de recherche du Québec - Santé. Grant # 281660"

4. Please expand the acronym “FRQS” (as indicated in your financial disclosure) so that it states the name of your funders in full.

Reviewers' comments:

Reviewer's Responses to Questions

**Comments to the Author**

1. Is the manuscript technically sound, and do the data support the conclusions?

Reviewer #1: Yes

Reviewer #2: Partly

2. Has the statistical analysis been performed appropriately and rigorously? 

Reviewer #1: Yes

Reviewer #2: Yes

3. Have the authors made all data underlying the findings in their manuscript fully available?

Reviewer #1: Yes

Reviewer #2: Yes

4. Is the manuscript presented in an intelligible fashion and written in standard English?

Reviewer #1: Yes

Reviewer #2: Yes

5. Review Comments to the Author

Reviewer #1: Because the existing mouse models of neurofibroma and MPNST do not recapitulate the key clinical and molecular features of the corresponding human tumors, the authors aimed to establish a model the recapitulated the human disease clinically and molecularly. Therefore they revisited the NP cis model. The paper is interesting well written and clear. The Authors also discussed the current state of NF mouse models, highlighting model relationship to human tumors.

Reviewer #2: In this manuscript Plante et al re-characterized the CisNf1+/-;P53+/- (NPcis) mouse.

Consistent with the published data, the NPcis mouse developed malignant peripheral nerve sheath tumor (GEM-PNST). The authors also observed lesions reminiscent of a neurofibroma in a peripheral nerve, a cutaneous neurofibroma, and para-spinal neurofibroma on rare occasions in

NPcis mice by a systematic histological analysis. They also observed that 10% of the mice developed GEM-PNSTs spontaneously. More interestingly, about half of the sciatic nerves from NPcis mice developed plexiform neurofibromas (PNFs) within 1-6 months when intentionally injured. Thus,this might provided a procedure to turn the widely used NPcis sarcoma model into a model recapitulating PNF. The authors have provided extensive of histology/immunohistochemistry data. Overall, this is a very well organized manuscript. Most of the data are convincing. However, some of the following concerns diminished my enthusiasm for publication at current version:

1. Nf-/- Schwann cells are the pathological cells for PNFs. In the NPcis nerve injury model, is there loss of heterozygosity (LOH)? Is there any difference between the mice with PNF and without PNF in the level of LOH? What about the environment change, especially the macrophages .

2. Is there any histological difference on PNFs developed by longitudinal or cross section procedure?

3. Some of the NPcis mice also develop brain tumors. This can cause mouse death too. It is important to be included in the characteristics.

6. PLOS authors have the option to publish the peer review history of their article (what does this mean?). If published, this will include your full peer review and any attached files.

Reviewer #1: No

Reviewer #2: No

---

## [Author Response · Author response to Decision Letter 0]

1 Mar 2024

OBJECT : Resubmission of Plante et al. PONE-D-23-19723

Dear Editors,

Thank you for the constructive comments and opportunity to address the editorial and reviewer`s comments on our manuscript entitled: ``Revisiting the NPcis mouse model: a new tool to model plexiform neurofibroma``. Please find below the point-by-point response.

EDITORIAL COMMENTS

We now use the PLoS ONE template

Thank you for the suggestion.

"Fonds de recherche du Québec - Santé. Grant # 281660"

The funder had no role, therefore thank you for amending the funder role in the online submission form as ""The funders had no role in study design, data collection and analysis, decision to publish, or preparation of the manuscript."" 

4. Please expand the acronym “FRQS” (as indicated in your financial disclosure) so that it states the name of your funders in full. This information should be included in your cover letter; we will change the online submission form on your behalf.

FRQS stands for ``Fonds de Recherche du Québec – Santé``. Thank you for amending it in the online submission form. 

We moved any ethic statement to the Method section.

We now follow the PLoS ONE template. The list of Supporting Information is now found at the end of the manuscript.

We fully revised the reference list.

REVIEWERS` COMMENTS

Reviewer #1: Because the existing mouse models of neurofibroma and MPNST do not recapitulate the key clinical and molecular features of the corresponding human tumors, the authors aimed to establish a model that recapitulated the human disease clinically and molecularly. Therefore, they revisited the NP cis model. The paper is interesting, well written and clear. The Authors also discussed the current state of NF mouse models, highlighting model relationship to human tumors.

We thank the reviewer for the enthusiasm about our manuscript.

Reviewer #2: In this manuscript Plante et al re-characterized the CisNf1+/-;P53+/- (NPcis) mouse. Consistent with the published data, the NPcis mouse developed malignant peripheral nerve sheath tumor (GEM-PNST). The authors also observed lesions reminiscent of a neurofibroma in a peripheral nerve, a cutaneous neurofibroma, and para-spinal neurofibroma on rare occasions in NPcis mice by a systematic histological analysis. They also observed that 10% of the mice developed GEM-PNSTs spontaneously. More interestingly, about half of the sciatic nerves from NPcis mice developed plexiform neurofibromas (PNFs) within 1-6 months when intentionally injured. Thus, this might provide a procedure to turn the widely used NPcis sarcoma model into a model recapitulating PNF. The authors have provided extensive of histology/immunohistochemistry data. Overall, this is a very well-organized manuscript. Most of the data are convincing. However, some of the following concerns diminished my enthusiasm for publication at current version:

Again, we thank the reviewer for the enthusiasm about our manuscript.

1. Nf-/- Schwann cells are the pathological cells for PNFs. In the NPcis nerve injury model, is there loss of heterozygosity (LOH)? Is there any difference between the mice with PNF and without PNF in the level of LOH? What about the environment change, especially the macrophages.

All excellent points. Ideally, we would have performed DNA sequencing. Unfortunately, the only material output from this study is FFPE slides.

-To determine whether or not there is LOH, we did the following two experiments which are featured in a brand-new Fig. 7 along with S6 and S7 Fig:

a) We performed immunostaining using NF1 and p53 antibodies on normal sciatic nerves (no injury) from wild-type mice (n=8), injury-induced sciatic nerves from NPcis mice that develop pNF (n=12), injury-induced sciatic nerves from NPcis mice that did not develop (n=9) pNF and spontaneous sarcoma from NPcis mice (n=23) “Fig 7a”. Then, the staining intensity was scored and output as a bar graph representing the percentage of tissues assigned to each staining intensity level. The results indicate that NF1 and p53 expression are largely preserved in non-malignant tissues whereas it is lowly expressed in malignant tissues as expected “Fig 7b”. 

b) We also performed qPCR using Nf1 and p53 specific primers on normal sciatic nerves (no injury) from wild-type mice (n=2), injury-induced sciatic nerves from NPcis mice that develop pNF (n=2), injury-induced sciatic nerves from NPcis mice that did not develop (n=2) pNF and spontaneous sarcoma from NPcis mice (n=2). Again, the results indicate that Nf1 and p53 expression are largely preserved in non-malignant tissues. More precisely, Nf1 expression is not significantly different between NPcis nerves and pNF but it is for p53. As expected, both Nf1 and p53 expression is significantly decreased in sarcoma compared to normal nerve from NPcis mice “Fig 7c”. Thus, it suggests that loss of heterozygosity of Nf1 and p53 did not took place in injury-induced pNF although alternative strategies to lower p53 expression may exist.

-To determine any changes in macrophages, we performed IHC using Iba1 antibodies on injury-induced sciatic nerves from NPcis mice that develop pNF (n=7), injury-induced sciatic nerves from NPcis mice that did not develop (n=8) pNF. As expected we found positive staining in the injury-induced pNF and minimal staining in the injury-induced sciatic nerves that did not develop pNF. This new result is now presented as S5 Fig.

2. Is there any histological difference on PNFs developed by longitudinal or cross section procedure?

We revisited the H&E from injury-induced neurofibroma from the ``needle`` and the ``cut`` to evaluate any difference in the histologies. This comparison is now shown in S4 Fig.

3. Some of the NPcis mice also develop brain tumors. This can cause mouse death too. It is important to be included in the characteristics.

It is true that NPcis develop a plethora of manifestations, including brain tumors. However, we strictly focused on the benign manifestations of the skin (cNF) and peripheral nervous system (pNF) in this study.

---

## [Decision Letter · Decision Letter 1]

11 Mar 2024

Revisiting the NPcis mouse model: a new tool to model plexiform neurofibroma

PONE-D-23-19723R1

Dear Dr. Brosseau,

We’re pleased to inform you that your manuscript has been judged scientifically suitable for publication and will be formally accepted for publication once it meets all outstanding technical requirements.

Kind regards,

Giulio Piluso, M.Sc.

Academic Editor

PLOS ONE

Additional Editor Comments (optional):

Reviewers' comments:

Reviewer's Responses to Questions

**Comments to the Author**

1. If the authors have adequately addressed your comments raised in a previous round of review and you feel that this manuscript is now acceptable for publication, you may indicate that here to bypass the “Comments to the Author” section, enter your conflict of interest statement in the “Confidential to Editor” section, and submit your "Accept" recommendation.

Reviewer #2: All comments have been addressed

2. Is the manuscript technically sound, and do the data support the conclusions?

Reviewer #2: Yes

3. Has the statistical analysis been performed appropriately and rigorously? 

Reviewer #2: I Don't Know

4. Have the authors made all data underlying the findings in their manuscript fully available?

Reviewer #2: Yes

5. Is the manuscript presented in an intelligible fashion and written in standard English?

Reviewer #2: Yes

6. Review Comments to the Author

Reviewer #2: The authors have addressed all of my concerns and the current version is now acceptable for publication.

7. PLOS authors have the option to publish the peer review history of their article (what does this mean?). If published, this will include your full peer review and any attached files.

Reviewer #2: No

---

## [Editor Report · Acceptance letter]

22 May 2024

PONE-D-23-19723R1 

PLOS ONE

Dear Dr. Brosseau, 

I'm pleased to inform you that your manuscript has been deemed suitable for publication in PLOS ONE. Congratulations! Your manuscript is now being handed over to our production team.

Kind regards, 

on behalf of

Professor Giulio Piluso 

Academic Editor

PLOS ONE